

# Usage of normalized soil moisture for improving the performance of rainfall thresholds for landslides along transportation corridors

Leila Rahimikhameneh[1], Abraham Alvarez Reyna[1], Jack Montgomery[1], and Frances O'Donnell[1]

[1]Department of Civil Engineering and Environment, Auburn University, Alabama, 36849, United States
*Correspondence to*: Leila Rahimikhameneh (lzr0043@auburn.com)

**Abstract.** Landslides along transportation corridors pose significant risks to infrastructure and public safety, necessitating accurate prediction and mitigation strategies. Many early warning systems for landslides are based on rainfall thresholds derived from historical data that distinguish landslide triggering from non-triggering events. However, it is widely recognized

that antecedent moisture conditions have a major impact on the likelihood of a particular rainfall event leading to a landslide. We aim to improve existing rainfall thresholds for landslides along highways by incorporating antecedent soil moisture conditions. The landslide inventory was compiled using data from inclinometers at suspected landslide sites and from landslide reports following major storm events along Alabama highways. This inventory was combined with precipitation data from the National Oceanic and Atmospheric Administration (NOAA) and soil moisture data from NASA's Soil Moisture Active Passive

(SMAP) satellite. We explored the accuracy of rainfall thresholds from previous studies for forecasting landslides along the highways of Alabama. Additionally, we investigated the potential of reducing the number of non-landslide events that exceed the thresholds (false positives) by utilizing soil moisture data derived from SMAP. This study demonstrates that sites with multiple inclinometers in a landslide region produce more robust datasets compared to those with a single inclinometer, enabling more effective differentiation between landslide and non-landslide events. Furthermore, using normalized soil

moisture in the development of rainfall thresholds shows potential for reducing false positives, as approximately 75 percent of the false positive cases in this study occurred when the soil moisture was at or below average conditions. Our proposed normalized soil moisture-dependent thresholds will support decision-making systems by enabling users to weigh the tradeoffs between potential false alarms and missed alarms, depending on the relative cost or risk of each for a given project. The findings will aid transportation authorities and civil engineers in making informed decisions about possible interventions or

preventative maintenance in the future.

## 1 Introduction

Landslides are frequent geohazards in many parts of the world. Transportation corridors like highways, railways, and tunnels are particularly vulnerable to landslides due to slope modification during construction and the potential for construction on pre-existing landslides. Comprehensive regional and global landslide databases are scarce, though a conservative estimate

indicates that around 11.7% of all landslides in a global database of non-seismically triggered events between 2004 and 2016



impacted road networks (Taylor et al., 2020). Cost estimates for landslides along transportation corridors typically include only direct expenses related to repairs, but landslides also lead to large indirect costs associated with traffic disruptions and road closures (Klose, 2015; Knights et al., 2020). Winter et al. (2016) stated that although landslide-affected transportation corridors rarely result in large fatalities, the social and economic impacts can be significant. These include delays and detours
on transportation networks, and the disruption of access for remote communities to services, markets, employment, healthcare, education, and social activities. The range of financial losses attributed to landslides is considerable, with estimates spanning from $400 million in 1971 to $2.5 billion in 2019 in the United States (Mirus et al., 2020). Minor landslides, though less documented, make up 96% of the events impacting roads and railways in Switzerland, resulting in an annual cost of CHF 6 million (USD 6.5 million) (Voumard et al., 2018). The impact of landslide-induced damage on transportation corridors can be
severe and long-lasting, especially when a strategic transportation route is affected. In such cases, the indirect costs (or consequential economic impacts) can be as large as the direct costs (Winter et al., 2016).

Rainfall is one of the most common triggers of landslides (Santangelo et al., 2023; Cepeda et al., 2010) and many warning systems for rainfall-triggered landslides rely on case history-based thresholds to determine if a given rainfall event is likely to lead to a landslide or not (Conrad et al., 2021). These thresholds are commonly developed by analyzing landslide databases
from past rainfall events, with many studies using the power-law equation proposed by Caine (1980) to separate events that triggered a landslide from those that did not. Guzzetti et al. (2007, 2008) compiled internationally developed thresholds prior to 2008, highlighting the key rainfall variables used in various studies to establish rainfall thresholds. Most studies utilized some combination of cumulated rainfall from an event (E), rainfall intensity (I), and rainfall event duration (D). The most widely applied threshold combinations include intensity-duration (I-D), cumulated rainfall-duration (E-D), and cumulated
rainfall-intensity (E-I), typically represented on semi-logarithmic, logarithmic, or Cartesian coordinate systems. A review of 115 thresholds developed between 2008 and 2016 (Segoni et al., 2018a) revealed that nearly 50% of the defined thresholds were based on I-D relationships, 16% were based on cumulated rainfall, and 27% relied on antecedent rainfall. The remaining thresholds fell into other categories (Segoni et al., 2018a).

Rainfall thresholds can be categorized based on their geographical extent as either global (e.g., Kirschbaum et al., 2010, 2015),
national (e.g., Lin et al., 2021; Millán-Arancibia and Lavado-Casimiro, 2023; Uwihirwe et al., 2020; Baum and Godt, 2010; Mirus et al., 2020), regional (e.g., Valenzuela et al., 2018, Roccati et al., 2020, Martelloni et al., 2012), or local (D'Ippolito et al., 2023). Limited regional and local studies have specifically focused on rainfall thresholds along transportation corridors. For example, Mandal and Sarkar (2021) divided a 54.8 km-long highway into four sections and established four rainfall intensity (I) and event duration (D) thresholds in a landslide-prone region of the Himalayas. Ray et al. (2010) defined a
threshold based solely on in situ and remotely sensed soil moisture for the Highway 50 corridor in the Sierra Nevada Mountains, California, USA, covering an area of 616 km². Other studies focused on mechanisms of landslide occurrence along highways without introducing thresholds (Fayaz et al., 2022; Sepúlvedaet al., 2023; Zhao et al., 2024). Abraham et al. (2021) used a two-dimensional Bayesian approach for landslide occurrence in Idukki, a hilly area in the Western Ghats of the Indian



Peninsula, where landslides triggered by heavy rainfall frequently disrupt the transportation network. Mirus et al. (2018a)
investigated landslides along the Seattle-Everett railway corridor and established thresholds based on the relationship between
3-day cumulative rainfall and 1-day antecedent soil saturation. Mirus et al. (2018b) had extensive monitoring data from the
railway that allowed for accurate analysis of antecedent rainfall and soil moisture conditions during the critical period leading
up to the failure. Unlike railroads, roads and highways are often less rigorously monitored, resulting in limited data on the
precise timing of landslide occurrences. The lack of temporal precision presents a significant challenge in defining reliable
thresholds for landslide prediction.

Bogaard and Greco (2018) and Segoni et al. (2018b) highlighted the absence of thorough hydro-meteorological analysis in
empirically based rainfall I-D thresholds for landslide initiation. Rainfall serves as the final "push" for initiating landslides,
while other factors such as soil moisture, infiltration, and storage and drainage capacity play vital roles (Gain et al., 2022).
Recent studies have employed hydro-meteorological thresholds, which differ from traditional intensity–duration (I–D) plots
by incorporating not only rainfall variables, such as cumulative rainfall and maximum intensity, but also soil moisture or
volumetric water content (e.g., Marino et al., 2020, Oorthuis et al., 2023).  Lazzari et al. (2020) employed two regression
models to analyse landslide occurrences, focusing on saturation degrees below and above 0.7 to identify the most critical
conditions for landslide initiation. However, their study only focused on landslide events and did not consider non-landslide
events. Wicki et al. (2020) utilized a soil moisture model based on soil hydrological properties and found that the success of
thresholds depends significantly on the distance between the measurement station, where the soil hydrological properties are
derived, and the landslide location. Due to the inherent uncertainty and spatial variability in soil moisture models, especially
in poorly instrumented regions, applying thresholds can lead to significant inaccuracies in predicting landslide initiation.

Remote sensing-based soil moisture datasets have become increasingly common in landslide studies as they allow for
measurements over a much larger area than instrument-based measurements (Brocca et al., 2012; Rodríguez-Fernández et al.,
2017; Skulovich and Gentine, 2023; Stanley et al., 2021). Zhuo et al. (2019) revealed that using the remotely sensed soil
moisture product from the ESA Climate Change Initiative (CCI-SM) showed that more than 80% of landslides happened when
soil moisture was in the top half of the wetness range. However, Yang et al. (2023) found that the use of remotely sensed soil
moisture data did not significantly enhance the performance of rainfall thresholds in Jiangjia Gully (China), primarily due to
its coarse spatial resolution. Abancó et al. (2024) utilized root-zone soil moisture data from SMAP L4 to assess landslide
susceptibility. However, they found that in tropical regions, the critical layer for landslide triggering during the wet season is
the unsaturated layer beneath the root zone, which cannot be captured using remotely sensed data. These studies highlight the
challenges of using remotely sensed products, as their effectiveness can vary across different regions depending on topography
and climate.

Our study focuses on evaluating rainfall and soil moisture thresholds for rainfall-triggered landslides along highways in
Alabama, where previous landslides have caused significant damage to roadways and disruption to traffic (Montgomery et al.,
2019; Knights et al., 2020). The objectives of this work are to: (a) determine which rainfall thresholds can be used to evaluate



landslides along highways in Alabama; and (b) explore how incorporating normalized soil moisture into the threshold may improve agreement with a focus on reducing false positives. We are not aware of any previous studies that have evaluated the effectiveness of rainfall thresholds in Alabama or surrounding states. For this study, we used landslide data obtained from two

sources: (1) inclinometer data collected at unstable sites along highways by the Alabama Department of Transportation over 20 years (2001 to 2021) and (2) reports of landslides along highways following federally declared disasters in Alabama over six years (2009 to 2015) as documented by Knights et al. (2020). The use of inclinometer readings to create a landslide inventory is advantageous as it allows for a clear delineation between landslide events and non-landslide events based on measured movements. A preliminary version of this inventory was presented by Rahimikhameneh et al. (2024), who studied

nine locations with similar deformation profiles, but this has been greatly expanded and refined in this study. We utilize the daily precipitation data from the CPC Unified Gauge-Based Analysis of Daily Precipitation (Xie et al., 2007) provided by the National Oceanic and Atmospheric Administration (NOAA) and NASA's Soil Moisture Active Passive Level 4 (SMAP-L4) data for soil moisture measurements (Reichle et al., 2018). Rather than using the volumetric water content directly, we chose to normalize the data using the average value at each location over our study period (2015 to 2021) to allow for a consistent

metric of moisture conditions across all the sites. We compared our inventory to previously developed I-D thresholds and found that the threshold proposed by Godt et al. (2006) is successful in predicting approximately 90% of the landslides from the inclinometer-based landslide inventory, but with a large number of false positives. When comparing the false positives and true positives, we found that the false positives tended to have drier than average conditions as measured by the normalized soil moisture and that an integrated threshold utilizing the normalized soil moisture was able to reduce the number of false

positives, while maintaining relatively similar accuracy for the true positive cases. Future work is needed to evaluate how the proposed thresholds perform over a longer time and to compare them against other landslide inventories, particularly for road networks with medium-range instrumentation such as inclinometers. The proposed method offers a means to incorporate regional soil moisture conditions into landslide thresholds for transportation networks.

## 2 Landslide Inventory

In this study, we utilized a landslide inventory derived from inclinometer readings and records of landslides triggered by major storm events along Alabama highways. The inclinometer datasets included quarterly readings from areas previously identified as having stability issues and captured events that may not have caused enough damage to be documented in reports or newspapers. We used movement thresholds to classify the measured displacements into distinct categories of landslide and non-landslide events, as discussed in the next section. This landslide inventory from inclinometers comprised 87 landslides,

48 of which occurred after March 31, 2015, aligning with the availability of soil moisture data from SMAP.

In addition to inclinometer data, we utilized the landslide inventory compiled by Knights et al. (2020) for landslides along Alabama highways. Specifically, we used landslides that were reported to the Federal Highway Administration (FHWA) to request federal funding for repairs between 2009 and 2015. Each landslide was attributed to a specific federally declared



disaster, which allowed for a precise definition of the triggering event for the landslide. This inventory did not provide
information on non-landslides, as only damaged sites were reported. The landslide inventory compiled by Knights et al. (2020)
includes 164 landslides that occurred between 2009 and 2015, 64 of which occurred after March 31, 2015, aligning with the
availability of SMAP data.

## 2.1 Inclinometer Processing

An inclinometer monitors deformation perpendicular to the axis of the casing, providing measurements of subsurface
horizontal deformation. The most common analysis for inclinometer data involves plotting the relative shape of the casing
compared to its initial condition. These cumulative lateral deformation plots are commonly used to identify potential shear
zones (Machan and Bennett, 2008; Stark and Choi, 2008). ALDOT utilized biaxial inclinometers at the sites considered in this
study, which provide measurements in both the A- and B-directions with a reading interval of 2 ft (0.61 m). Typically, the A-
direction aligns with the direction of the maximum displacement, as was the case at all of the inclinometers in our study. We
extracted the displacements along the inclinometers using DigiPro2 software (Durham Geo Slope Indicator) and created CSV
files for subsequent processing. We generated cumulative lateral deformation versus depth plots for all dates in a single profile
for each A-direction and B-direction using Python 3.10 and the following libraries: pandas, NumPy, os, and matplotlib.

The cumulative lateral deformation profiles for each inclinometer reading were processed to identify potentially erroneous
readings and to identify the depth of the top of the shear zone at each site. Inclinometers without any movement events during
the monitoring period were removed from the inventory. We filtered erroneous readings by removing readings with significant
changes in displacement (>2.5 mm) between two consecutive readings at the bottom of the casing and readings with spikes in
displacement at a single depth without corresponding movements at other depths. After removing erroneous readings, each
deformation profile was manually reviewed to identify potential shear zones. Quantifying other potential sources of error in
inclinometer readings can be more challenging. Mikkelsen (2003) reviewed errors in inclinometer measurements and estimated
that the random error in inclinometer readings is approximately ±0.16 mm for an individual reading. This random error
accumulates at a constant rate over the entire length of the casing. Therefore, the accumulated random error for a 30-meter
casing with readings taken every 0.5 meters would be approximately +1.24 mm at the top of the casing. Random errors cannot
be detected and removed, but understanding their magnitude offers a potential threshold to separate potential noise in the
inclinometer data from true movements.

Figure 1 presents examples of cumulative lateral deformation profiles from three different inclinometers used in this study.
Profile A is an example of a site with no movement and was not included in the analysis as it did not represent a location that
was susceptible to landslides. Profile B reveals two distinct shear zones, suggesting multiple failure planes, whereas Profile C
displays significant displacement along a single shear zone and substantial ground deformation. For the profile types
represented by B and C, displacement values were extracted from the top of the shallowest shear zone in each reading.
Additionally, inclinometers with deformations extending to the bottom of the casing (indicating the casing does not extend



past the unstable mass) were excluded from the inventory. These were excluded because movements originating beyond the bottom of the casing make it difficult to reliably determine the location and characteristics of the actual shear zone.

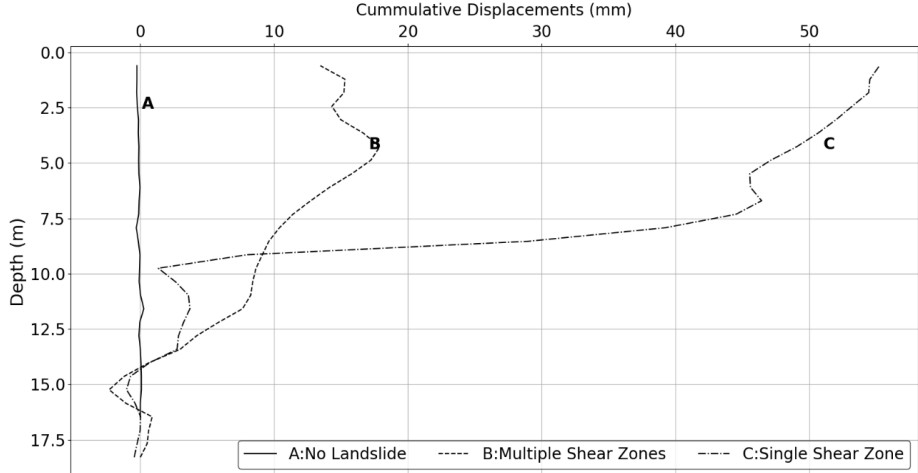

**Figure 1. Examples of cumulative lateral displacement profiles of inclinometer casings relative to the initial position.**

Following this screening process, the final inventory included 56 inclinometers with identifiable shear zones distributed across 19 sites. The list of inclinometers with the latitude and longitude, site name, and depth of shear zone is shown in Table A1 (Appendix). We also categorized each site based on the stratigraphy near the shear zone as either weathered shale, interbedded sands and clays, or high-plasticity clay using the geologic map from Szabo et al. (1988). Of the 19 sites, seven are located in

geologic units consisting primarily of high-plasticity clay, nine had were failures in weathered shale layers, and three were in units with interbedded sand and clay layers (Table A1). The inventory includes six sites with only a single inclinometer and 13 sites with multiple inclinometers. Figure 2 shows the landslide locations from the two inventories in the study region: the inclinometer-based inventory (this study) and the landslide inventory compiled by Knights et al. (2020).

The extracted displacements from the inclinometers were used to determine if enough movement had occurred to be classified

as a landslide event. Rahimikhameneh et al., (2024) examined histograms of recorded displacements and defined non-landslide events as monitoring periods with a change in displacement smaller than 1 mm between two consecutive readings and landslide events as a monitoring period with a change in displacement larger than 5 mm between two consecutive readings. These thresholds are also used in this study. Events with displacements between 1 and 5 mm pose a challenge as they could indicate small landslide events or measurement errors. As no clear method exists to distinguish between these possibilities, such

readings were excluded from this analysis to focus only on confirmed landslide and non-landslide events.



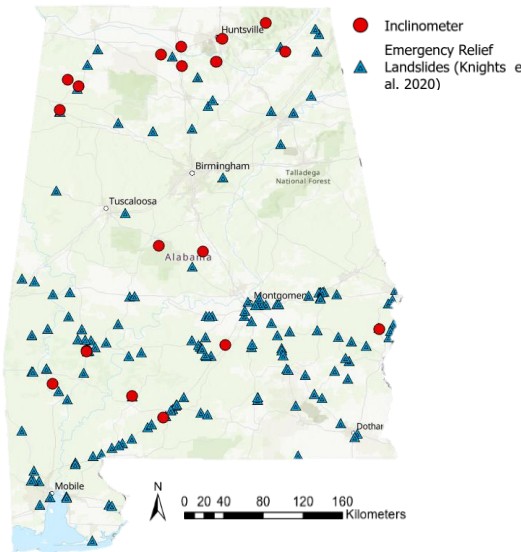

**Figure 2. Spatial distribution of landslide sites from the processed inclinometer and landslide inventory compiled by Knights et al., 2020 (Base map showing topography from ESRI 2025).**

## 2.2 Precipitation and Soil Moisture Processing

We used Python (v3.11) and ArcGIS Pro (v3.0, ESRI) to process the precipitation and soil moisture data. The precipitation dataset used in this study is the CPC Unified Gauge-Based Analysis of Daily Precipitation over CONUS data provided by the NOAA PSL, Boulder, Colorado (https://psl.noaa.gov). The product has a spatial resolution of 28-km by 28-km. Daily
precipitation from CPC NOAA was grouped into discrete storm events by using a rainy-day threshold of 1 mm, as it is commonly used for I-D threshold development (Leonarduzzi et al., 2017). To calculate rainfall intensity, the cumulative rainfall for each storm event was calculated and then divided by the number of days the storm persisted. Almost all precipitation in Alabama falls as rain and was treated accordingly in the analysis. We did not normalize the rainfall intensity by mean annual precipitation, since annual precipitation remains relatively consistent across Alabama, ranging from 127 to 152 cm per year.

Soil moisture data from NASA's SMAP satellite (https://appeears.earthdatacloud.nasa.gov) were used for this project, which has a 9-km by 9-km resolution. We specifically used the root zone moisture from the Level 4 product (Reichle et al., 2018), as this was found to be the most applicable to shallow landslides by Marino et al. (2020). Given this shallow depth and coarse resolution, the root zone soil moisture is considered as a regional indicator of average wetness in this study and not a site-specific measure of soil moisture, nor a proxy for the matric suction at the depth of the shear zone.

An example of the processed data is shown in Figure 3 for landslide sites on Alabama Highway 69 (AL-69) and Alabama Highway 5 (AL-5). The inclinometer readings indicating landslide events (> 5 mm of displacement) are shown as stars. Figure 3 shows that these landslide events are most prevalent when soil moisture levels are higher than average. In other words,



rainfall of the same intensity that previously did not trigger landslides could induce landslides if the soil moisture condition is sufficiently elevated. The two sites exhibit different ranges and average values of soil moisture, highlighting the challenge of comparing datasets between sites. To address this challenge, we used normalized soil moisture on the first day of the storm in prediction thresholds. Normalized soil moisture is calculated as the soil moisture measured by SMAP at the site divided by the average soil moisture of that site between 2015 and 2021. This is used as an index representing the moisture conditions near the slide area relative to the average conditions at the same location. A normalized soil moisture of 1.0 represents an average condition for that location, while a value less than 1.0 is drier than average and greater than 1.0 is wetter than average. This normalization was performed in order to compare soil moisture conditions across the state using a single, consistent metric.

## 2.3 Relating landslides and storm events

As illustrated in Figure 3, many storm events occurred over each inclinometer reading period. If the reading interval showed a change in displacement of less than 1 mm, all storms in that reading interval were considered non-landslide events, as no detectable movement occurred during that reading interval. When a landslide event was detected between two inclinometer readings, we assumed the event with the largest cumulative precipitation during the reading period triggered the landslide. The other storm events during that reading interval were excluded from the analysis, as there is uncertainty in which event (or combination of events) caused the movement. For this reason, our database has more reliable estimates of non-triggering storm events, but uncertainty in the magnitude of precipitation for triggering events from the inclinometer data.

As previously mentioned, most sites were instrumented with multiple inclinometers. Instead of analysing each inclinometer independently, inclinometers at the same site were grouped to determine a failure status for the entire site. If a landslide was detected by at least one inclinometer (change in displacement > 5 mm) at a monitoring site, the entire site was considered to have experienced failure during that reading interval, designating the associated storm as a triggering event. A non-landslide event was defined as such if the change in displacement for the entire group of inclinometers at a site was less than 1 mm during that reading interval. For sites with only a single inclinometer, we used the same threshold to identify triggering events (movements greater than 5 mm). We excluded the non-triggering events from single inclinometer sites in our analysis, as we observed that it was common at multiple inclinometer sites for one inclinometer to exceed the threshold (indicating the site had failed) while others did not. This leads to uncertainty in non-triggering events for sites with only a single inclinometer.




(a)

(b)

**Figure 3. Time series of inclinometer displacement (red star indicates landslide and open event indicate non-landslide), daily rainfall (purple bars), and soil moisture (SMAP L4, blue line) for (a) AL-69 Inclinometer 13002 and (b) AL-5 Inclinometer 13002A.**



## 2.4 Assessing threshold performance

The performance of each threshold was evaluated using a confusion matrix or contingency table. A true positive (TP) corresponds to an I-D pair, representing rainfall intensity (I) and duration (D), that exceeds the threshold and is associated with an observed landslide event. A false positive (FP) refers to an I-D pair that exceeds the threshold without a corresponding

landslide occurrence, representing a false alarm. A false negative (FN) occurs when the I-D pair falls below the threshold, yet a landslide is recorded, indicating that the threshold failed to capture a hazardous event. A true negative (TN) is an I-D pair below the threshold for which no landslide is observed, correctly identifying a non-landslide event (Piciullo et al., 2017).

To quantify the performance of each threshold, the performance metrics used by Piciullo et al. (2017) were applied. The Probability of Detection (POD), also known as the true positive rate (TPR) (Eq. 3), represents the proportion of actual positive

cases correctly identified. The Probability of False Detection (POFD), also called the false positive rate (FPR) (Eq. 4), refers to the proportion of non-landslides incorrectly predicted as landslides. The Probability of False Alarm (POFA), also known as the false discovery rate (Eq. 5), reflects how good the system is at avoiding unnecessary alarms. Lastly, the Hanssen and Kuipers Skill Score (HK) (Hanssen and Kuipers 1965), also referred to as the true skill statistics (Eq. 6), reflects how well the threshold distinguishes between positive and negative cases:


$$TPR = \frac{TP}{TP+FN} \tag{3}$$

$$FPR = \frac{FP}{TP+TN} \tag{4}$$

$$POFA = \frac{FP}{TP+FP} \tag{5}$$

$$HK = \frac{TP}{TP+FN} - \frac{FP}{FP+TN} \tag{6}$$

TPR measures the proportion of actual positive cases correctly identified and ranges from 0 (no positives detected) to 1 (all

positives detected), with higher values indicating better sensitivity. In contrast, FPR quantifies the proportion of negative cases incorrectly classified as positive, and lower FPR values are desirable as they reflect fewer false alarms. POFA represents the proportion of predicted positive events that are actually false; thus, lower POFA values indicate greater precision in the model's predictions. The HK score, defined as the difference between TPR and FPR, evaluates the model's ability to distinguish between positive and negative cases. HK scores closer to 1 suggest strong discriminatory power, while values near 0 imply

that the model performs no better than random guessing.

## 2.5 Example of data processing: AL-219

The processing steps described above are illustrated in this section using the landslide site on Alabama Highway 219 (AL-219) as an example. This site is located within the Gordo formation of the Tuscaloosa group, near the boundary with the Coker formation (Szabo et al., 1988). The Gordo formation mainly consists of cross-bedded sand, gravelly sand, and lenticular clay

beds, with the lower part dominated by gravelly sand containing chert and quartz pebbles. The Coker formation is composed of micaceous sand and clay with some gravel layers containing quartz and chert pebbles. Figure 4 illustrates the distribution



of the four inclinometers at the site. It is common practice to install multiple inclinometers at varying distances across a landslide site to monitor ground movement in different zones of the affected area. Consequently, these inclinometers often show non-uniform patterns of movement despite all being within the same landslide.


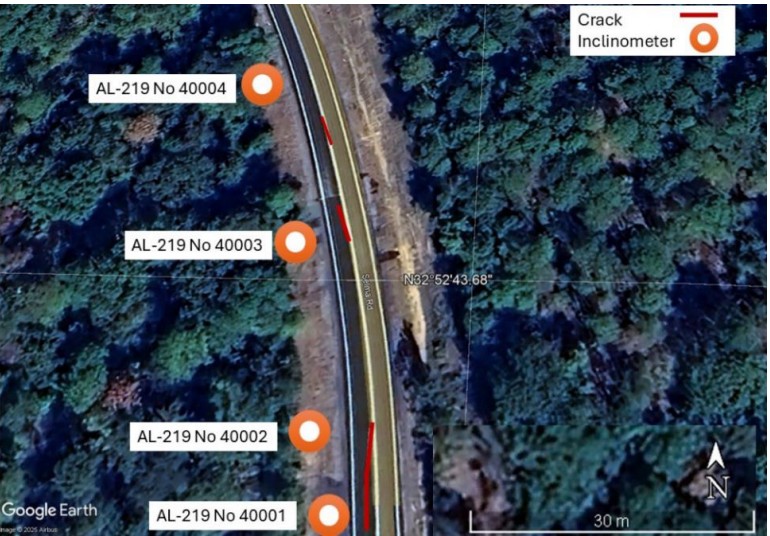

**Figure 4. Location of four inclinometers at AL-219 site in Centerville and cracks caused by landslide, AL (Base map from © Google Earth 2025).**

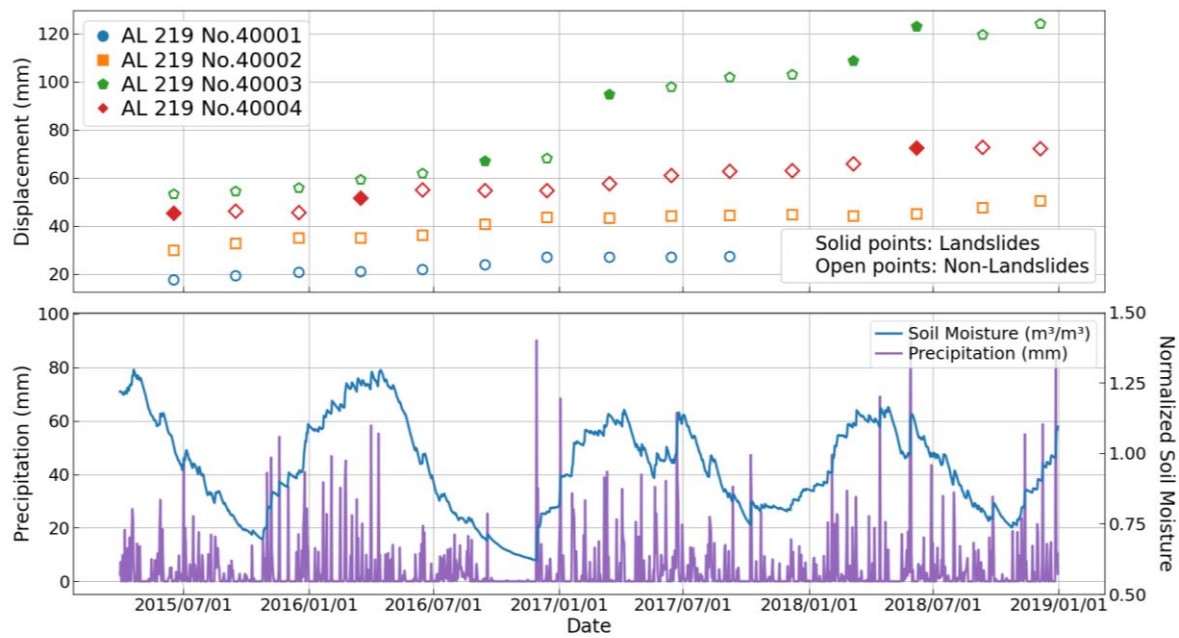

**Figure 5. Time series showing landslide events and displacements of four inclinometers of a monitoring site (solid event indicates landslide and open event indicate non-landslide) integrating with precipitation and normalized soil moisture.**



The displacement recorded by inclinometers at AL-219, along with the rainfall and normalized soil moisture, is shown in Figure 5. Inclinometers AL-219 40003 and AL-219 40004 recorded landslide events during the study period, but not always in the same reading interval. For example, AL-219 40003 detected movement on 2017-01-23 while no movement was observed in the other inclinometers. Both AL-219 40003 and AL-219 40004 recorded movement on 2018-05-31, indicating activity at
different locations of the same landslide. AL-219 40001 and AL-219 40002 did not show any movement during the study period and therefore were not used in the analysis. These observations emphasize the importance of using a cluster of displacement recordings to monitor and capture movements across an affected area during a landslide event.

## 3 Results

After the processing steps described above, our final inclinometer-based inventory includes 87 landslide events, representing
storm events that caused movements greater than 5 mm, and 905 non-landslide events, representing storm events that caused movements less than 1 mm, based on inclinometer data. Additionally, 164 landslide events were extracted from the Knight et al. (2020) database. We have made the data for these landslides available in the DesignSafe data repository (Rahimikhameneh et al. 2025). The subset of the inclinometer data for which SMAP data is available includes 48 landslide events and 363 non-landslide events, while this subset of the Knight et al. (2020) inventory contains 64 landslide events.

### 3.1 Comparison with Existing Triggering Thresholds

We conducted a comparison of storms that triggered landslides, utilizing data from the processed inclinometers and landslide inventory compiled by Knights et al. (2020), against previously established thresholds by Godt et al. (2006); Guzzetti et al. (2008); and Marino et al. (2020) to assess their applicability to unstable sites along highways in Alabama. It is important to note that the role of normalized soil moisture in improving the prediction was not taken into account in this step, and the events
analyzed belong to the entire inventory spanning from 2001 to 2021. Effects of normalized soil moisture are investigated in the next section using a subset of the inventory, as the SMAP data were only available after 2015.

To evaluate the performance of different rainfall thresholds in predicting landslide events, we compared three empirical models to the inclinometer-based inventory: Godt et al. (2006), Marino et al. (2020), and Guzzetti et al. (2008), using confusion matrix-based metrics as shown in Table 1. As shown in Figure 6a and Figure 6b, the Godt et al. (2006) provides the best overall
balance, with a high TPR (0.85), low FPR (0.11), and the highest HK of 0.75, indicating strong discriminatory power. Marino's model is less conservative, yielding the lowest FPR (0.03) and a relatively low POFA (0.35), but at the cost of lower sensitivity (TPR = 0.64) due to the large number of landslide points that fall below the threshold. In contrast, the Guzzetti model achieves the highest sensitivity (TPR = 0.94) but with a high FPR (0.34) and POFA (0.79), indicating a tendency to overpredict landslide occurrence relative to our inventory.





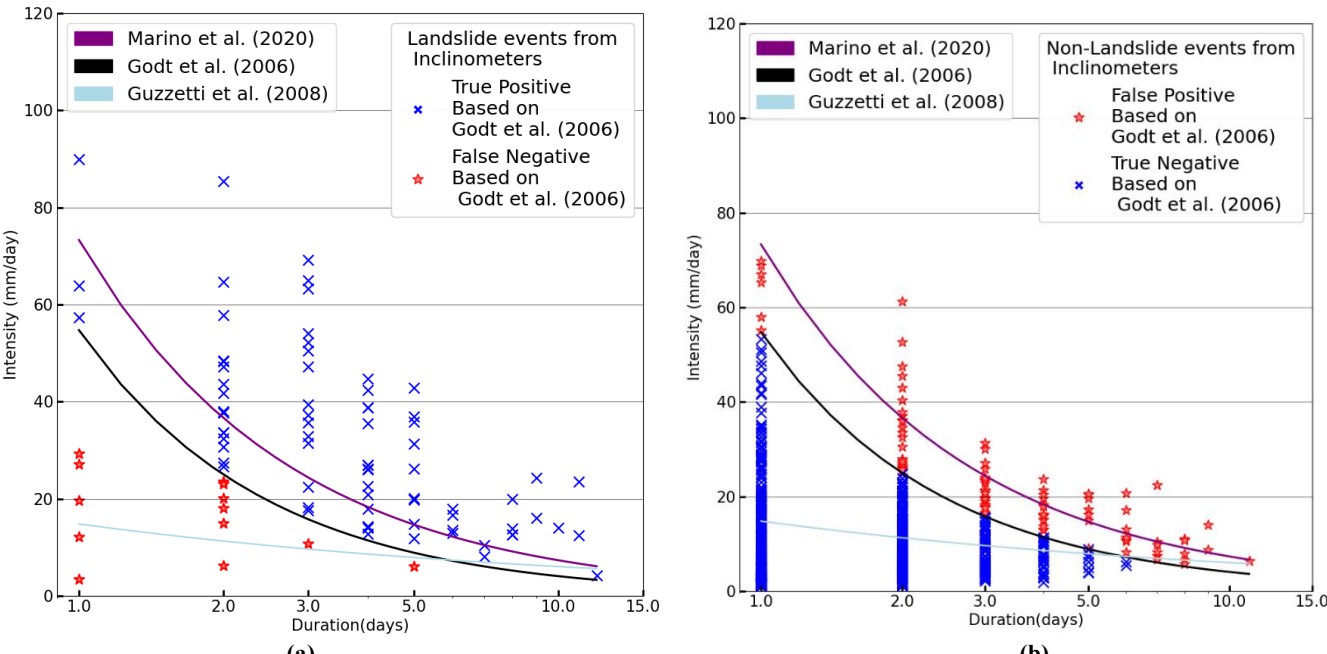

**Figure 6. Comparison of the (a) landslide events and (b) non-landslide events with previously developed rainfall thresholds for the inclinometer-based inventory from this study.**

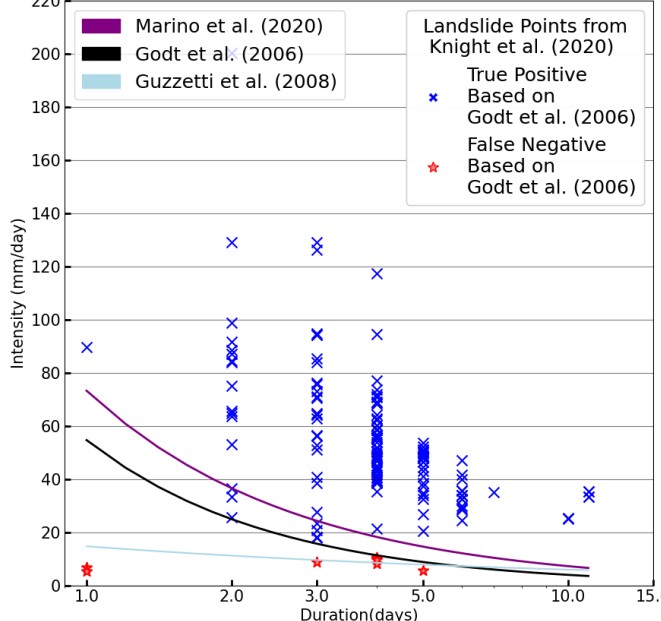

**Figure 7. Comparison of the landslide events with previously developed rainfall thresholds for the landslide inventory compiled by Knights et al. (2020).**





**Table 1.** **Comparison of performance metrics for previously developed thresholds for the inclinometer inventory**

| Threshold model | TP | FP | FN | TN | TPR | FPR | POFA | HK |
|---|---|---|---|---|---|---|---|---|
| Godt et al. (2006) | 74 | 95 | 13 | 810 | 0.851 | 0.105 | 0.562 | 0.764 |
| Marino et al. (2020) | 56 | 30 | 31 | 875 | 0.644 | 0.033 | 0.350 | 0.610 |
| Guzzetti et al. (2008) | 82 | 308 | 5 | 597 | 0.943 | 0341 | 0.790 | 0.602 |

We also compared all three thresholds with the landslide inventory compiled by Knights et al. (2020), as shown in Figure 7.
The Godt et al. (2006) threshold correctly identified 156 of the 164 (95.1%) landslide events with only eight false negatives.
The Guzzetti et al. (2008) and Marino et al. (2020) thresholds had similar performances with the Guzzetti et al. (2008) being
the most conservative and therefore having the fewest number of false negatives. We did not compute statistics for this
inventory as it does not include any non-landslide points. These comparisons demonstrate that the Godt et al. (2008) threshold
provides the most balanced fit for our inventory, and we will focus on this threshold in the remaining sections.

**3.2 Landslide Events Considering Soil Moisture**

The large number of false positives in the previous section led us to examine whether including the normalized soil moisture
along with the precipitation data could help separate events that were more or less likely to cause a landslide. Both landslide-
triggering and non-landslide-triggering events were assigned a normalized soil moisture value by taking the soil moisture on
the first day of the storm event and normalizing it by the average over the study period (2015 – 2021). The normalized soil
moisture values for the storm events ranged from 0.2 to 2.15. We chose to bin the values in symmetric bins with an interval of
0.1, and the outermost bins (below 0.75 and above 1.25) were extended to ensure sufficient data events in each bin. Normalized
soil moistures between 0.95 and 1.05 represent approximately average conditions, values below 0.95 represent drier-than-
average conditions, and values greater than 1.05 represent wetter-than-average conditions. The inclinometer-based inventory
is shown with these bins in Figure 8 and the Knights et al. (2020) inventory is shown in Figure 9. The number of events has
been reduced compared to Figures 6 and 7, as SMAP data are only available after March 31, 2015.

Figure 8(a) shows the results for the landslide events binned by the normalized soil moisture measured on the first day of the
storm. Three landslide events from the inclinometer-based inventory fell below the threshold established by Godt et al. (2006),
indicating false negatives. These three points had either average (0.95-1.05) or above-average (greater than 1.05) normalized
soil moisture values. The comparison for the non-landslide events (less than 1 mm) is shown in Figure 8(b). The Godt et al.
(2006) threshold had 44 false positives out of 363 total non-landslide events (12%). Approximately 75% of these false positive
events had a moisture content at or below average (less than 1.05). The inventory compiled by Knights et al. (2020) is shown
in Figure 9, with all landslides falling above the threshold. Only the landslides from the December 2015 storms in the Knights
et al. (2020) inventory overlapped with the availability of SMAP data and all the landslides from these storms had at or above
average moisture conditions.





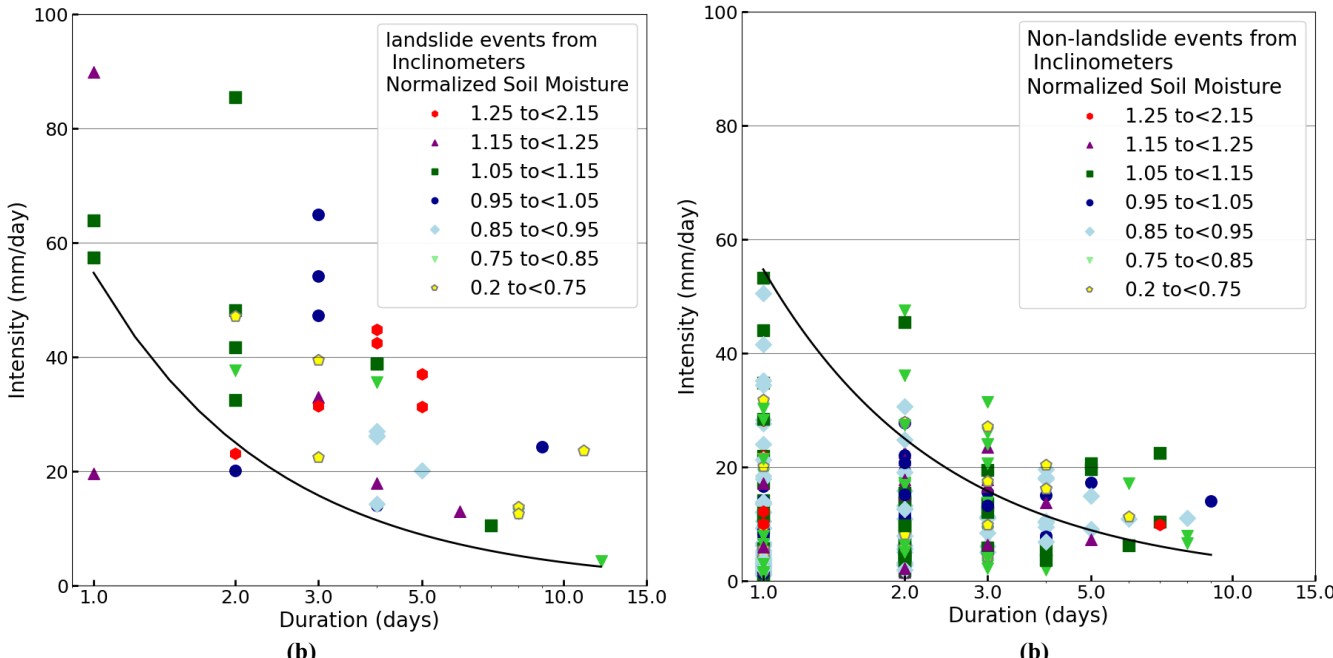

**Figure 8. Comparison of the (a) landslide events and (b) non-landslide events grouped by normalized soil moisture with previously developed rainfall thresholds for the inclinometer database from this study compared to Godt et al. (2006).**

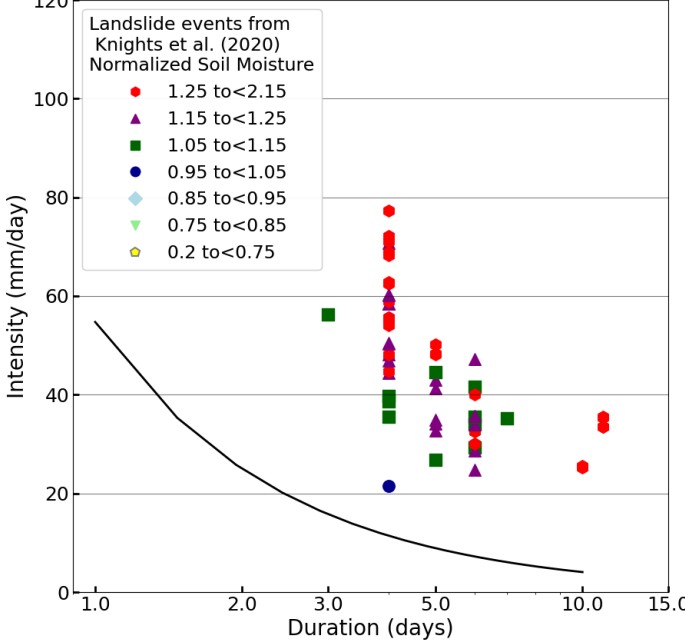

**Figure 9. Comparison of the landslide events grouped by normalized soil moisture with previously developed rainfall thresholds for the landslide inventory compiled by Knights et al. (2020).**




Histograms of normalized soil moisture for true positive (landslide events above the Godt et al. 2006 threshold) and false positive events (non-landslide events above the Godt et al. 2006 threshold) from the inclinometer-based inventory and DDIR-based inventory compiled by Knights et al. (2020) are shown in Figure 10. Figure 10a shows that 60% of the true positive events for the inclinometer-based inventory occurred at times of average or above average moisture conditions. More than 65% of the false positive events (Figure 10b) had normalized soil moisture values lower than 0.95 (drier than average conditions). Only 2.3% of the false positive events have normalized soil moisture values greater than 1.25 compared with 13%

**Figure 10. Histogram of normalized soil moisture values based on the threshold developed by Godt et al. (2006), showing: (a) true positive events, (b) false positive events from the inclinometer database used in this study, and (c) true positive events from the landslide inventory compiled by Knights et al. (2020).**





of landslide events. For the DDIR-based inventory, all events were above the Godt et al. (2006) threshold, and the average normalized soil moisture was 1.3 (Figure 10c). Taken together, Figures 8 -10 suggest that threshold curves that incorporate moisture conditions may offer improved capability to distinguish between storms that are more or less likely to trigger a
landslide by separating true and false positive events. This is further explored in the next section.

## 4 Proposed approach to integrate normalized soil moisture into landslide thresholds

The intensity-duration (I-D) threshold proposed by Godt et al. (2006) effectively predicts the landslides in our inventory (approximately 90%) but also shows many false positives (Figure 8b). As previously discussed, all the non-landslide events in our inventory are from sites with a history of instability, so these false positives cannot be attributed to a lack of susceptibility.
Our analysis in the previous section showed that more than half of the false-positive events occurred when normalized soil moisture was drier than average which indicates that incorporating normalized soil moisture may improve the threshold by reducing the number of false positives. To investigate this further, we combined the events from the two landslide inventories (inclinometer-based and DDIR-based inventories) and grouped the events into five classes based on normalized soil moisture values: three classes ranging from 0.75 to 1.05 in 0.1 intervals, and two additional classes for values below 0.75 or above 1.05,
respectively. This classification follows the same approach used previously for soil moisture binning, but we chose to combine all of the above-average soil moisture values (NSM > 1.05) into a single bin, as we did not find any trend among the wetter-than-average points.

To develop our normalized soil moisture (NSM)-dependent thresholds, we used the threshold proposed by Godt et al. (2006) as a baseline. The Godt et al. (2006) threshold is defined by the power law shown in Eq. (1):

$$I_{Godt} = 82.73D^{-1.13} \qquad (1)$$

where I is the rainfall intensity in mm per hour, and D is the duration in hours. For consistency with our inventory, we modified Eq. 1 to convert the rainfall intensity to mm per day and duration to days and added a new NSM-dependent fitting parameter (α) in Eq. (2):

$$I_{NSM\ Thresholds} = \alpha \times 54.73D^{-1.13} \qquad (2)$$

where I is the rainfall intensity in mm per day, D is the duration in days, and α is a fitting parameter that depends on the normalized soil moisture (NSM). Two different NSM-dependent thresholds were fit based on prioritizing either the reduction of false positives (NSM-dependent Threshold A) or the reduction of false negatives (NSM-dependent Threshold B). The selected α value for each NSM bin is shown in Table 2.




**Table 2. Equations of lines and corresponding scaling factors for normalized soil moisture (NSM)-dependent thresholds A, B.**

| | Threshold (NSM<0.75) | Threshold (0.75<NSM<0.85) | Threshold (0.85<NSM<0.95) | Threshold (0.95<NSM<1.05) | Threshold (NSM>1.05) |
|---|---|---|---|---|---|
| NSM-Dependent Threshold A | $I = 102.90D^{-1.13}$ | $I = 82.10D^{-1.13}$ | $I = 67.32D^{-1.13}$ | $I = 61.30D^{-1.13}$ | $I = 57.47D^{-1.13}$ |
| | $\alpha = 1.88$ | $\alpha = 1.50$ | $\alpha = 1.23$ | $\alpha = 1.12$ | $\alpha = 1.05$ |
| NSM-Dependent Threshold B | $I = 102.90D^{-1.13}$ | $I = 82.10D^{-1.13}$ | $I = 67.32D^{-1.13}$ | $I = 61.30D^{-1.13}$ | $I = 49.80D^{-1.13}$ |
| | $a = 1.41$ | $a = 1.28$ | $a = 1.23$ | $a = 1.12$ | $a = 0.91$ |

Figure 11 compares the NSM-Dependent Threshold A and the landslide and non-landslide events, alongside the original threshold proposed by Godt et al. (2006). As expected, higher normalized soil moisture levels correspond to lower rainfall intensity thresholds, indicating that landslides may be triggered by lower rainfall intensities over the same durations when the normalized soil moisture is higher. Table 3 shows the summary of threshold performance using the same metrics as before (Table 1). The NSM-Dependent Threshold A matches the sensitivity (TPR) of the Godt threshold reasonably well (0.952 vs.

0.971), indicating a comparable ability to identify actual landslide events. However, it reduces the FPR from 0.121 to 0.077 and lowers the POFA from 0.301 to 0.219. This improvement is also reflected in the HK score, which increases from 0.850 to 0.875, suggesting enhanced overall discriminative performance. This analysis demonstrates that the NSM-Dependent Threshold A provides improved performance over the threshold proposed by Godt et al. (2006) by reducing both the false positive rate and the probability of false alarm, alongside enhancement in discriminative capability.

To further investigate the trade-off between sensitivity and false alarms, NSM-Dependent Threshold B was developed with the objective to reduce the number of false negatives and thereby enhance the sensitivity of the threshold model, while accepting a moderate increase in false positives compared with Threshold A. The threshold lines are illustrated in Figure 12, where the line corresponding to the class of NSM > 1.05 lies below the Godt et al. (2006) threshold, indicating that under wetter than average conditions, smaller storms may be sufficient to trigger instability. The performance metrics for Threshold

B are also shown in Table 3. When compared to the Godt et al. (2006) threshold, Threshold B demonstrates an improvement in detection capability, achieving a higher TPR (0.981 vs. 0.971) and a lower number of false negatives (FN = 2 vs. 3). Although this improvement in sensitivity is accompanied by a slight increase in FPR compared with Threshold A, there are still fewer false positives than the original relationship. These results suggest that Threshold B may be more suitable for applications where the cost of missed events outweighs the consequences of increased false alarms.

**Table 3. Comparison of performance metrics for Godt et al. (2006) and the NSM-Dependent Threshold A, B.**

| Threshold model | TP | FP | FN | TN | TPR | FPR | POFA | HK |
|---|---|---|---|---|---|---|---|---|
| Godt et al. (2006) | 102 | 44 | 3 | 319 | 0.971 | 0.121 | 0.301 | 0.850 |
| NSM-Dependent Threshold A | 100 | 28 | 5 | 335 | 0.952 | 0.077 | 0.219 | 0.875 |
| NSM-Dependent Threshold B | 103 | 38 | 2 | 325 | 0.981 | 0.105 | 0.269 | 0.876 |



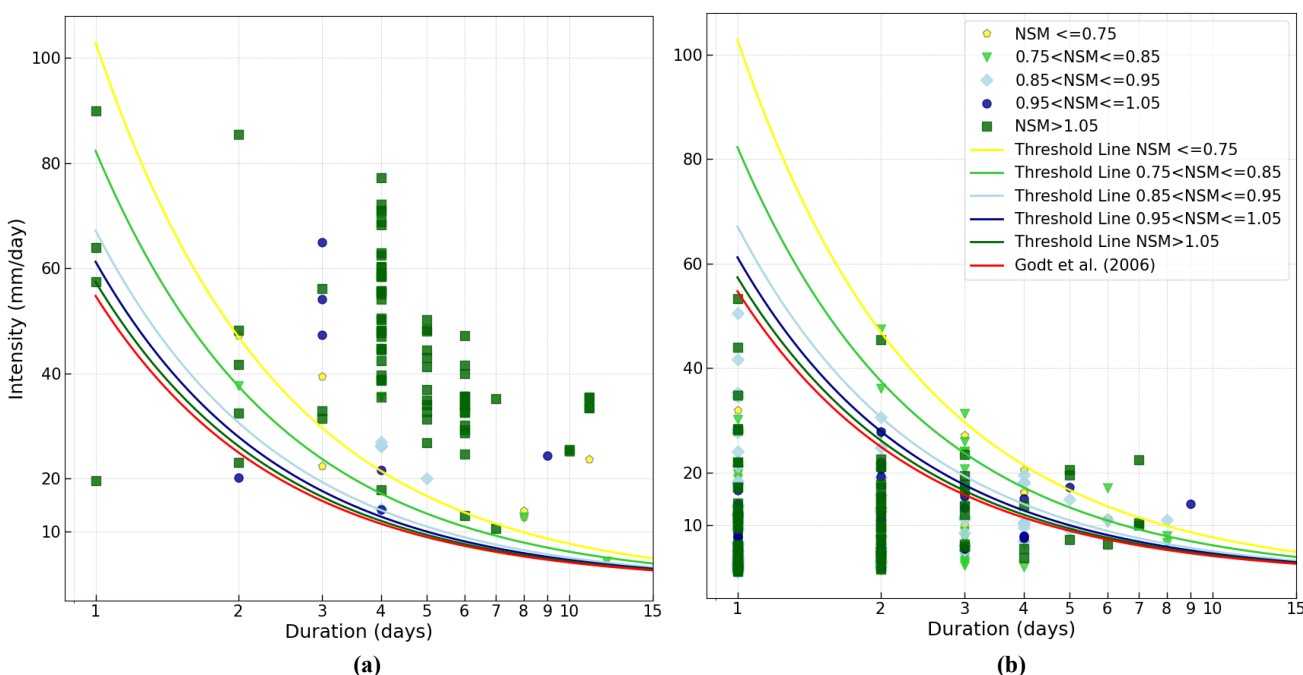

**Figure 11. NSM-Dependent I-D threshold A lines for the rainfall-triggering landslide fitted to events within each normalized soil moisture classification for (a) landslide events and (b) non-landslide events.**

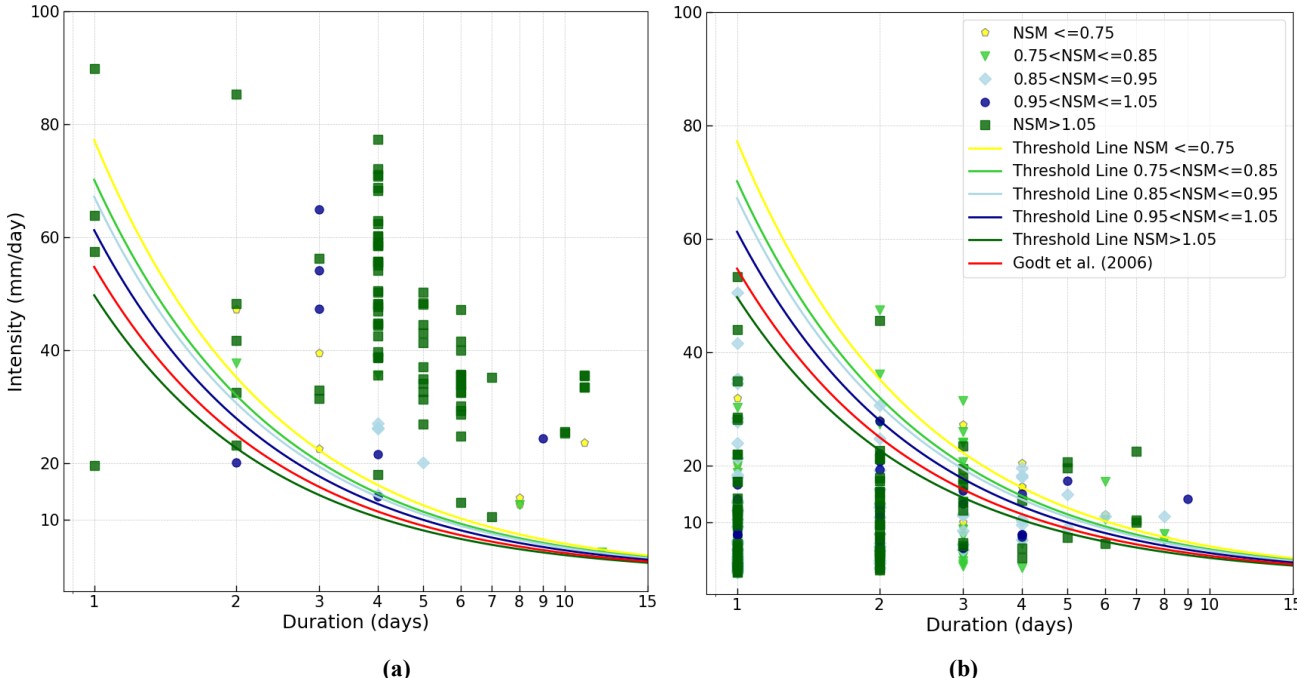

**Figure 12. NSM-Dependent I-D threshold B lines for the rainfall-triggering landslide fitted to events within each normalized soil moisture classification (Reducing 1 False negatives) for (a) landslide events and (b) non-landslide events.**



## 5 Discussion

Our study has demonstrated that normalized soil moisture based on the SMAP L4 Root Zone moisture is one potential tool to reduce false positives from rainfall-based thresholds. We developed and compared two NSM-Dependent Thresholds (A and B) with the baseline threshold proposed by Godt et al. (2006) and found an improvement in the ability to correctly identify true positives, minimize false alarms, and improve overall classification performance. Threshold B emerged as the most sensitive model, achieving the highest true positive rate (TPR = 0.981) and the lowest number of false negatives (FN = 2). This makes it the best option for scenarios where missing a positive case could have severe consequences. Furthermore, Threshold B also demonstrated the highest Hanssen-Kuipers (HK) score (0.876), reflecting the best overall classification performance among other thresholds. Threshold A had the lowest number of false positives and would be most useful in situations where false alarms need to be reduced.

Previous studies have reported mixed results regarding the use of remotely sensed soil moisture in landslide prediction. While some showed limited improvement due to spatial resolution issues, others noted that landslides often occur under extremely wet conditions. In this study, we addressed these challenges by normalizing soil moisture data, which reduced site-specific variability and enabled more consistent threshold development. Unlike raw soil moisture values, which can vary widely across regions, normalized soil moisture effectively captures relative wetness levels that contribute to slope failure risk. These results suggest that normalized soil moisture may be a more robust and transferable index for identifying landslide-prone conditions. However, this finding needs to be confirmed with other data sources and in other regions.

Our analysis considers a relatively limited inventory collected from unstable areas across Alabama. While the initial results are promising, the relatively small number of events and the region-specific characteristics of the inventory, such as local geology, hydrological behavior, and precipitation patterns, limit the generalizability of the findings. Therefore, further studies are needed in areas with different geological, hydrological, and climatic conditions to assess the broader applicability of NSM-dependent thresholds. This study also only focused on movements at unstable sites along highways and therefore does not apply to potential triggering of first-time landslides at sites without a history of movement.

One limitation of this study is the uncertainty in the exact timing of landslide events, as the inclinometer data were collected only on a quarterly basis. This time gap makes it difficult to precisely link landslides to specific rainfall events and their corresponding normalized soil moisture conditions, which may reduce the accuracy of threshold calibration. A more detailed inventory with known landslide occurrence dates would improve the evaluation and validation of the proposed thresholds.

## 6 Conclusion

This study created a new inventory of landslide and non-landslide events based on inclinometer readings collected at sites with unstable slopes around Alabama. Extensive processing was done on the inclinometer data to eliminate erroneous or unreliable readings and to extract changes in displacement at the shear zones. After processing, landslide events were defined by a change



in displacement exceeding 5 mm between two inclinometer readings, while non-landslide events were identified by changes in displacement less than 1 mm. Readings falling between these two limits were not considered in the analysis as they could
not be definitively categorized as either landslide or non-landslide with the available information. The landslide inventory compiled by Knights et al. (2020) was also included in the analysis to increase the number of landslide points.

The inclinometer-based inventory was compared with measured precipitation data from NOAA and soil moisture from NASA's SMAP Level 4 dataset. To allow for a uniform metric for comparison across the sites, the actual soil moisture values were normalized by the average measured at that site over the study period to create a normalized soil moisture (NSM), which
serves as an index of the average moisture conditions in the vicinity of the landslide site. A comparison of data at selected sites showed that landslides tended to occur during periods of higher NSM. The two inventories were compared with previously developed rainfall thresholds and the threshold proposed by Godt et al. (2006) was found to accurately predict approximately 92% of the landslide events across the two inventories. Using the Godt et al. (2006) threshold to predict landslide events resulted in approximately 12% false positives, with an average normalized soil moisture (NSM) of 0.896 indicating conditions
drier than the overall average. Examining the full inventory showed that 75% of the inclinometer-based landslides and 100% of the DDIR-based inventory (Knights et al. 2020) had NSM values above 1, indicating wetter than average conditions. This suggests that incorporating NSM into the threshold formulation could be a promising approach to reduce false positives.

The events from the two inventories were combined and binned into five NSM classes. Threshold lines were fit to each class under the constraint of maintaining or improving classification performance relative to the Godt et al. (2006) threshold. NSM-
Dependent Threshold A focused on reducing false positives at the expense of increasing the number of false negatives, while Threshold B prioritized sensitivity by reducing false negatives in wetter conditions at the expense of a controlled increase in false positives. NSM-Dependent Threshold B demonstrated the highest sensitivity, with a true positive rate (TPR) of 0.981 and the lowest number of false negatives (FN = 2), making it particularly suitable for scenarios where missing a positive case could lead to critical consequences. Threshold A had the lowest false positive rate (FPR = 0.077) and the lowest probability of false alarm (POFA = 0.219), making it more appropriate for applications where minimizing false alarms is essential, such
as automated alert systems or contexts prone to alert fatigue. Both NSM-Dependent Thresholds outperformed the original Godt et al. (2006) model, underscoring the value of incorporating normalized soil moisture to enhance the reliability and specificity of rainfall-induced landslide early warning systems.

The proposed approach enhances traditional rainfall-based landslide thresholds by incorporating normalized soil moisture, allowing for improved accuracy and reduced false positives. Its effectiveness relies on sufficient spatial variability in soil
moisture to enable classification into distinct wetness groups and the fitting of tailored threshold lines. When these conditions are met, the method offers a more reliable, location-sensitive model for landslide prediction along highways. Further work is needed to determine the effectiveness of the proposed thresholds in other regions.



*Code and data availability*. The landslide inventories are available through the DesignSafe Data Repository
(https://doi.org/10.17603/ds2-xs04-th22).

*Competing interests*. The authors declare that they have no conflict of interest.

*Acknowledgments.* This material is based upon work funded by the Alabama Department of Transportation under grant number 931-054 and the National Science Foundation under grant number CMMI 2047402. Inclinometer data was provided by Brannon McDonald (ALDOT). Any opinions, findings, conclusions, or recommendations are those of the author(s) and do not necessarily reflect the views of ALDOT or the National Science Foundation.

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



Table A1: Description of the locations used in the study

| Number | Highway | Number | Site Name | Longitude | Latitude | Shear Zone Depth | Stratigraphy |
|---|---|---|---|---|---|---|---|
| 1 | AL-219 | 40002 | Bibb-1 | 32.87811 | -87.1022 | 18 | Interbedded sand and clay |
| 2 | AL-219 | 40003 | Bibb-1 | 32.87811 | -87.1022 | 10 | Interbedded sand and clay |
| 3 | AL-219 | 40004 | Bibb-1 | 32.87811 | -87.1022 | 14 | Interbedded sand and clay |
| 4 | AL-22 | 11001 | Chilton | 32.83098 | -86.71118 | 15 | Interbedded sand and clay |
| 5 | AL-22 | 11003 | Chilton | 32.83098 | -86.71118 | 14 | Interbedded sand and clay |
| 6 | AL-5 | 13001A | Clarke-1 | 31.94514 | -87.7359 | 28 | High plasticity clay |
| 7 | AL-5 | 13002A | Clarke-1 | 31.94514 | -87.7359 | 16 | High plasticity clay |
| 8 | AL-69 | 13002 | Clarke-2 | 31.6586 | -88.033899 | 14 | High plasticity clay |
| 9 | US-43 | 13007 | Clarke-3 | 31.95097 | -87.74001 | 11 | High plasticity clay |
| 10 | I-65 | 18001 | Conecuh | 31.35986 | -87.066741 | **12** | High plasticity clay |
| 11 | AL-187 | 30007 | Franklin-1 | 34.35024 | -87.904297 | 10 | Weathered shale |
| 12 | AL-187 | 30008 | Franklin-1 | 34.35024 | -87.9043 | 10 | Weathered shale |
| 13 | AL-146 | 36008 | Jackson-1 | 34.85264 | -86.1637 | 10 | Weathered shale |
| 14 | AL-146 | 36017 | Jackson-1 | 34.85264 | -86.1637 | 15 | Weathered shale |
| 15 | AL-146 | 36018 | Jackson-1 | 34.85264 | -86.1637 | 14 | Weathered shale |
| 16 | AL-146 | 36018A | Jackson-1 | 34.85264 | -86.1637 | 13 | Weathered shale |
| 17 | AL-35 | 36019 | Jackson-2 | 34.59687 | -85.996392 | 19 | Weathered shale |
| 18 | AL-35 | 36020 | Jackson-2 | 34.59687 | -85.996392 | 16 | Weathered shale |
| 19 | I-65 | 42003 | Limestone | 34.6446 | -86.90638 | 11 | Weathered shale |
| 20 | I-65 | 43012 | Lowndes | 32.00358 | -86.523023 | 16 | High plasticity clay |
| 21 | I-65 | 43002 | Lowndes | 32.00358 | -86.523023 | 19 | High plasticity clay |
| 22 | I-65 | 43003 | Lowndes | 32.00358 | -86.523023 | 18 | High plasticity clay |
| 23 | I-65 | 43004 | Lowndes | 32.00358 | -86.523023 | 14 | High plasticity clay |
| 24 | US-431 | 45001 | Madison-1 | 34.71448 | -86.545817 | 11 | Weathered shale |
| 25 | US-431 | 45002 | Madison-1 | 34.71448 | -86.545817 | 11 | Weathered shale |
| 26 | US-43 | HAMB3 | Marion-1 | 34.08032 | -87.976327 | 20 | Interbedded sand and clay |





| 27 | US-43 | HAMB4 | Marion-1 | 34.08032 | -87.976327 | 38 | Interbedded sand and clay |
| 28 | US-43 | Marb-7 | Marion-2 | 34.29364 | -87.805732 | 10 | Weathered shale |
| 29 | AL-41 | 50001A | Monroe | 31.55234 | -87.336817 | 10 | High plasticity clay |
| 30 | I-65 | 52005A | Morgan-1 | 34.47104 | -86.898092 | 18 | Weathered shale |
| 31 | I-65 | 52006A | Morgan-1 | 34.47104 | -86.898092 | 10 | Weathered shale |
| 32 | I-65 | 52007 | Morgan-1 | 34.47104 | -86.898092 | 10 | Weathered shale |
| 33 | I-65 | 52008 | Morgan-1 | 34.47104 | -86.898092 | 42 | Weathered shale |
| 34 | I-65 | 52009 | Morgan-1 | 34.47104 | -86.898092 | 26 | Weathered shale |
| 35 | AL-24 | 52001 | Morgan-2 | 34.57404 | -87.084189 | 29 | Weathered shale |
| 36 | AL-24 | 52002 | Morgan-2 | 34.57404 | -87.084189 | 23 | Weathered shale |
| 37 | AL-24 | 52003 | Morgan-2 | 34.57404 | -87.084189 | 11 | Weathered shale |
| 38 | US-231 | 52018 | Morgan-4 | 34.51145 | -86.597974 | 48 | Weathered shale |
| 39 | US-231 | 520189 | Morgan-4 | 34.51145 | -86.597974 | 10 | Weathered shale |
| 40 | US-231 | 52020 | Morgan-4 | 34.51145 | -86.597974 | 26 | Weathered shale |
| 41 | US-431 | 57001 | Russell-1 | 32.14236 | -85.165491 | 10 | High plasticity clay |
| 42 | US-431 | 57002 | Russell-1 | 32.14236 | -85.165491 | 10 | High plasticity clay |
| 43 | US-431 | 57003 | Russell-1 | 32.14236 | -85.165491 | 10 | High plasticity clay |
| 44 | US-431 | 57004 | Russell-1 | 32.14236 | -85.165491 | 10 | High plasticity clay |
| 45 | US-431 | 57005 | Russell-1 | 32.14236 | -85.165491 | 22 | High plasticity clay |
| 46 | US-431 | 57009 | Russell-1 | 32.14236 | -85.165491 | 28 | High plasticity clay |
| 47 | US-431 | 57012 | Russell-1 | 32.14236 | -85.165491 | 20 | High plasticity clay |
| 48 | US-431 | 57014 | Russell-1 | 32.14236 | -85.165491 | 11 | High plasticity clay |
| 49 | US-431 | 57015 | Russell-1 | 32.14236 | -85.165491 | 10 | High plasticity clay |
| 50 | US-431 | 57016 | Russell-1 | 32.14236 | -85.165491 | 20 | High plasticity clay |
| 51 | US-431 | 57023 | Russell-1 | 32.14236 | -85.165491 | 10 | High plasticity clay |
| 52 | US-431 | 57017 | Russell-1 | 32.14236 | -85.165491 | 10 | High plasticity clay |
| 53 | US-431 | 57018 | Russell-1 | 32.14236 | -85.165491 | 66 | High plasticity clay |
| 54 | US-431 | 57022 | Russell-1 | 32.14236 | -85.165491 | 26 | High plasticity clay |
| 55 | US-431 | 57030 | Russell-1 | 32.14236 | -85.165491 | 10 | High plasticity clay |
| 56 | US-431 | 57031 | Russell-1 | 32.14236 | -85.165491 | 10 | High plasticity clay |