# Peer review of "Usage of normalized soil moisture for improving the performance of rainfall thresholds for landslides along transportation corridors"

_EGUsphere, 2025_

## Author Comment (AC2)

We are grateful to the editors and all three reviewers for taking the time to provide feedback on our manuscript. The comments and suggestions will greatly improve the overall quality, clarity, and rigor of the paper. We have responded to all reviewer comments and plan to add all revisions to our manuscript. Following the instructions from the editor, we have not made all revisions yet, but we will ensure that all revisions are clearly reflected in the updated text. Below, the original comments appear in bold roman typesetting, and our responses are provided in italic typesetting.

**Comments from reviewer 2**

**This study introduces the concept of incorporating normalized soil moisture (NSM) into rainfall-triggered landslide threshold models to enhance early warning performance. The integration of long-term inclinometer data, NOAA precipitation records, and NASA SMAP soil moisture observations provide a valuable multidisciplinary framework. However, from the standpoint of scientific rigor and generalizability, the manuscript still presents notable limitations in terms of data representativeness, statistical validation, spatial-scale consistency, and geotechnical mechanism interpretation. Major revision is required before the manuscript can be considered for publication. The authors are encouraged to strengthen the temporal resolution, address spatial resolution discrepancies, and provide quantitative uncertainty analyses to improve the reliability and applicability of the conclusions.**

*We thank the review for their thorough review of our manuscript. In the revised manuscript, we improve the discussion of the limitations to better qualify our conclusions and avoid overstating the applicability of our findings. We will also clarify that we did not intend to present generalizable relationships, but rather a transferable framework that can be applied to other regions where sufficient data is available. We recognize that our data has limitations both temporally and spatially, but we are unable to change these. We do plan to more clearly acknowledge these limitations in the revised manuscript.*

*We intend to address each of the specific comments that reviewer #2 has identified through revisions to the manuscript. Our responses are detailed below.*

1. **The manuscript frequently uses the first-person pronoun, which should be removed for formal scientific writing. For instance, "We aim to improve existing rainfall thresholds for landslides along highways by incorporating antecedent soil moisture conditions." should be revised to "This study aims to improve existing rainfall thresholds for landslides along highways by incorporating antecedent soil moisture conditions."**

*Response to Comment 1:* *We thank the reviewer for this suggestion. It is our understanding that both first-person and third-person constructions are acceptable within the journal, and we have tried to maintain a consistent style with other literature on this topic. We believe that this is more a matter of preference than a technical comment and so have respectfully decided to maintain our current style. We will, however, review the final manuscript carefully to ensure that our writing style is consistent and does not detract from the technical contributions of the manuscript.*

2. **The use of quarterly inclinometer readings results in low temporal resolution. This makes it impossible to accurately link specific rainfall events with landslide occurrences, severely weakening the temporal correspondence between rainfall–duration (I–D) thresholds and soil moisture response, and consequently reducing statistical correlation and causal interpretability.**

*Response to Comment 2:* *Thank you for this important comment. We acknowledge that the use of quarterly inclinometer readings results in low temporal resolution. This is, unfortunately, an inherent limitation of many landslide databases, where the exact timing of slope movement is rarely known with high precision. We agree that this weakens the statistical correlations between landslide events and specific storms and will clarify this limitation in the revised manuscript.*

3. **The manuscript does not clarify the training background or consistency evaluation of the operators involved in inclinometer data acquisition, which could introduce subjective bias.**

*Response to Comment 3:* *We agree that this information is important. In the revised manuscript, we will clarify that all inclinometer readings were collected using the same inclinometer probe by the same trained geologist who has more than 10 years of experience in landslide monitoring in Alabama.*

4. **The spatial resolution of the SMAP data (9 km × 9 km) is too coarse to represent local-scale topographic and soil moisture variability, limiting the applicability of the model in mountainous areas with small-scale landslides.**

*Response to Comment 4:* *Thank you for this comment. We agree that the 9 × 9 km spatial resolution of the SMAP dataset is relatively coarse and may not capture local-scale variations in topography or soil moisture. We selected products with coarser spatial resolution because they provide the temporal resolution needed to analyze and predict landslide behavior. We will add text to better explain this rationale at line 191. Nonetheless, SMAP was selected because it provides publicly available, continuous, spatially consistent, and long-term observations that are appropriate for regional-scale assessments where finer-*

scale moisture measurements are unavailable. The objective of this study was to develop a landslide prediction threshold that could be applied across Alabama. While collecting site-specific, ground-based soil moisture data may provide a more accurate threshold, it would severely limit the broader application of the threshold to locations where in situ soil moisture data is collected. In this study, the SMAP root-zone soil moisture product is intentionally used as a regional indicator of average wetness, not as a site-specific measurement or a proxy for matric suction at the depth of the shear zone. The manuscript explicitly states this distinction (Line 193-195). The use of normalized soil moisture further reduces bias associated with local variability by expressing moisture conditions relative to each grid cell's long-term average, making comparisons across the state more robust. We will clarify these points in the revised manuscript, and we will also note that future work could incorporate higher-resolution soil moisture datasets or site-specific field measurements to improve representation of local-scale hydrologic variability.

5. **The sample size and regional representativeness are limited. The dataset is small and highly localized, resulting in weak generalizability of the conclusions and making it difficult to substantiate the claim of a "transferable threshold."**

*Response to Comment 5: We agree that the term "transferable threshold" was not appropriate given the regional nature of our study. It was not our intention to create a generalizable threshold, but rather to present an integrated approach to develop thresholds when sufficient datasets are available. In the revised manuscript, we will instead use the phrase "transferable methodology" rather than "transferable threshold," as this more accurately reflects the intended contribution of the study. The approach we present can be applied elsewhere, but the specific threshold values should indeed be considered region-specific, and this will be clarified.*

6. **Figure 3 fails to clearly demonstrate the relationship between rainfall, soil moisture, and landslide occurrence. The graphical evidence does not convincingly support the authors' interpretation and should be clarified with enhanced visualization or statistical quantification.**

*Response to Comment 6: Our intention with these figures was to provide motivation and context for our approach and not to demonstrate statistical correlation between these values. We will revise the text to better describe the intention of these figures.*

7. **Geological and soil parameters were not quantitatively controlled. Although the manuscript classifies strata into three lithologic types, key physical parameters (e.g., permeability, cohesion, plasticity index) are not incorporated, which weakens the geotechnical basis of the proposed thresholds.**

*Response to Comment 7: We agree with the reviewer that our approach does not control for differences in geotechnical properties between the different sites. The purpose of the lithological classifications was to provide context for the types of geologic units that are susceptible to landslides in Alabama. We agree that developing site-specific geotechnical properties is important for analyzing single slope failures or designing repairs, but our aim is to provide a regional assessment of timing of movements at already unstable locations. We would see our thresholds as a compliment to geotechnical slope stability analyses and not a replacement. We will ensure this motivation is clearly described in the introduction of the revised manuscript.*

8. **The innovation is more phenomenological than mechanistic. The study focuses on statistical correlation without sufficient exploration of the underlying hydro-geotechnical processes that govern the observed trends, reducing the theoretical depth of innovation.**

**Response *to Comment 8:*** *We agree that the present study is primarily phenomenological in nature, as our focus is on developing empirical relationships between rainfall, soil moisture, and slope movements across a relatively large region. We believe this is consistent with the aims of the journal, which include regional analyses that document observable patterns, improve understanding in data-limited environments, and provide a transparent, citable foundation for future scientific development. Our study provides an initial, data-driven assessment that documents observable regional patterns in Alabama's highway slopes. The intention is not to replace mechanistic hydro-geotechnical models, but rather to establish an approach to provide regional-scale assessments of when instabilities are likely to occur. We will clarify this motivation in the introduction to our manuscript.*

9. **The applicability boundaries of the proposed approach are not explicitly discussed. This omission reduces the methodological rigor and limits understanding of the model's valid domain.**

*Response to Comment 9: Thank you for this comment. In the revised version, we will clarify the valid domain of the methodology, emphasizing that the developed thresholds are specific to monitored highway slopes in Alabama and reflect the region's geological, hydrologic, and climatic conditions. As noted in our response to Comment 5, the threshold values themselves are not intended to be transferable; rather, the contribution of the study lies in the transferable methodology used to integrate rainfall, soil moisture, and displacement records showing the empirical workflow can be applied in other regions.*

10. **The rainfall event classification is overly simplified, and the threshold selection may be too lenient, as it does not account for the independence of consecutive dry periods or short-duration, high-intensity rainfall events.**

*Response to Comment 10: We agree that the previous discussion of event classification was unclear. In this study, rainfall events were separated following the approach of Leonarduzzi et al. (2017), which defines an independent rainfall event as a sequence of consecutive days with more than 1 mm/d of precipitation. This 1 mm/d threshold ensures that annual rainfall totals are not unrealistically reduced and that multiday storm durations remain physically meaningful. Under this definition, a dry period of at least 24 hours (i.e., a day with < 1 mm of rainfall) indicates the end of one event and the beginning of another, meaning that consecutive storms separated by a full dry day are treated as separate events. Using this event-separation algorithm, all rainy days without a dry period in between were grouped into the same storm. For each identified storm, cumulative rainfall was computed by summing all daily precipitation within the event, the number of rainy days was taken as the event duration, and rainfall intensity was calculated as cumulative rainfall divided by event duration. This provided a consistent metric for rainfall intensity across all events. We acknowledge that this daily resolution method does not explicitly account for very short, high-intensity bursts or the independence of shorter dry periods, as more advanced approaches require sub-daily rainfall data. However, the availability of hourly meteorological data is limited to a small number of sites and time periods, so a threshold based on sub-daily rainfall data would have limited use as a predictive tool. We will ensure that the revised manuscript more clearly explains this approach and acknowledges the limitations inherent in using daily rainfall.*

11. **The discussion section focuses primarily on whether prediction accuracy improved, but lacks an in-depth analysis of data sources, model structure, and scale compatibility. A more comprehensive discussion of these factors is needed; substantial revision of this section is recommended.**

*Response to Comment 11: We agree that the discussion section should more explicitly address the limitations in the study identified in both this and other comments. In the revised manuscript, we will substantially expand the Discussion and Limitations sections to incorporate these elements.*

*Specifically, we will add:*

- ***Data source limitations**, including the spatial resolution of SMAP (9 × 9 km), the daily resolution of precipitation records, and the quarterly temporal resolution of inclinometer measurements.*

- **Model structure limitations**, *noting that the thresholds are derived from empirical relationships and do not explicitly consider the hydro-geomechanical processes or site-specific material properties (e.g., permeability, cohesion, shear strength) governing slope stability.*

- **Scale-compatibility considerations**, *clarifying how regional-scale hydrologic indicators (rainfall, SMAP soil moisture) interact with highly localized slope movements measured by inclinometers, the tradeoffs between temporal and spatial resolution in gridded data products, and why this mismatch affects interpretability and transferability.*

12. **Performance metrics are reported without confidence intervals or statistical significance testing. Consequently, the claimed improvements cannot be validated statistically. The authors should incorporate cross-validation or independent testing.**

*Response to Comment 12:* *Thank you for this suggestion. In response to this comment and Comment 4 from Reviewer 3, we computed receiver operating characteristic (ROC) curves and AUC values for all five thresholds (Figure R1). In addition, we evaluated the thresholds using the Matthews Correlation Coefficient (MCC) to assess classification performance under a fixed operational decision rule (Table R1). The results show that both NSM-dependent thresholds (A and B) achieve higher AUC values than the three existing thresholds, indicating improved overall discrimination ability. When evaluated using MCC, Thresholds A and B also outperform the Godt et al. model, reflecting a more favorable balance between true and false classifications by incorporating normalized soil moisture. While the AUC differences among the highest-performing thresholds are modest, the combined use of AUC and MCC demonstrates that the NSM-dependent thresholds provide both strong discrimination and improved operational performance. This combined evaluation framework will be reflected in the revised manuscript.*

**Table R1. Comparison of landslide threshold performance using area under the ROC curve (AUC), Matthews Correlation Coefficient (MCC), and confusion matrix components.**

| Method | AUC | MCC | TP | FP | TN | FN |
|---|---|---|---|---|---|---|
| Threshold A | 0.983 | 0.824 | 100 | 27 | 336 | 5 |
| Threshold B | 0.9852 | 0.805 | 103 | 36 | 327 | 2 |
| Godt et al. (2006) | 0.98 | 0.765 | 102 | 44 | 319 | 3 |
| Marino et al. (2020) | 0.980 | 0.843 | 94 | 15 | 348 | 11 |
| Guzzetti et al. (2008) | 0.968 | 0.549 | 104 | 121 | 242 | 1 |

[Figure]

**Figure R1. Receiver operating characteristic (ROC) comparison of five intensity–duration thresholds: the two NSM-dependent thresholds (A and B), the Godt et al. (2006)**

threshold (α = 1), the Marino et al. (2020) threshold, and the Guzzetti et al. (2008) threshold.

13. The normalization procedure may obscure extreme moisture conditions. Averaging across long periods can reduce contrast between very wet and very dry states, thereby weakening the detection of extreme antecedent conditions that critically influence landslide initiation.

*Response to Comment 13: We agree that our previous explanation of the normalization procedure was unclear. For applying the NSM-dependent thresholds, we did not perform any averaging but rather used the soil moisture value from the start of the storm event. We did normalize the value using the long-term average, but the same average was used to normalize all of the readings for a given location and therefore would not obscure extreme values. We will clarify this in the revised manuscript.*

---

## Author Comment (AC3)

We are grateful to the editors and all three reviewers for taking the time to provide feedback on our manuscript. The comments and suggestions will greatly improve the overall quality, clarity, and rigor of the paper. We have responded to all reviewer comments and plan to add all revisions to our manuscript. Following the instructions from the editor, we have not made these revisions yet, but we will ensure that all revisions are clearly reflected in the updated text. Below, the original comments appear in bold roman typesetting, and our responses are provided in italic typesetting.

**Comments from Reviewer 3:**

**The manuscript presents research on improving existing rainfall thresholds for landslide prediction along highways by incorporating antecedent soil moisture conditions. The authors established an inventory containing landslide and non-landslide events, precipitation data from NOAA, and soil moisture data from NASA's SMAP. Rainfall thresholds from the literature for landslide forecasting were examined using the inventory data. Furthermore, the research proposed incorporating normalized soil moisture into the development of rainfall thresholds, which shows potential for reducing false positives in prediction. The work is highly practical and will be of interest to practitioners in landslide assessment and management. However, the overall quality of the manuscript needs improvement before it can be accepted for publication.**

*Dear Dr. Yichuan Zhu,*

*Thank you very much for your thoughtful and constructive review of our manuscript. We appreciate your recognition of the practical significance of our work. We will carefully revise the manuscript to address all of the points you raised. Below, we address each of your comments individually and provide detailed responses to clarify all points raised.*

1. **In the Discussion section, the authors mention spatial resolution issues, which remain a significant concern for the current study. NASA's SMAP operates at a 9-km by 9-km resolution, while CONUS data has a spatial resolution of 28-km by 28-km. How do these resolutions align with the site-specific study presented in the manuscript?**

*Response to Comment 1: While the study used site-specific inclinometer data as the response variable (landslide occurrence) for threshold development, the objective of the research was to provide a predictive tool for landslides that is applicable to sites where in situ data are not available. Therefore, publicly available gridded datasets were used as explanatory variables for the thresholds. There is a tradeoff between spatial and temporal resolution with gridded data products, and we will add additional explanation of our rational for using products with relatively coarse resolution and more detail on how the products*

*were applied. We will also clarify in the introduction that we see our thresholds as a compliment to site-specific geotechnical slope stability analyses and not a replacement.*

2. **The soil moisture data from NASA's SMAP satellite may require calibration before incorporation into the working pipeline. Based on the reviewer's experience, systematic bias between SMAP and in-situ soil moisture monitoring can exhibit seasonal patterns. It would strengthen the manuscript if the authors could provide additional justification regarding this potential issue.**

*Response to Comment 2: We agree that for site-specific calibration would be ideal for developing a mechanistic understanding of the relationship between soil moisture, matric suction, and landslide triggering. However, our objective in this study was to develop an empirical threshold for landslide warning systems that is applicable at sites where in situ data are not available. Within this empirical threshold, uncalibrated SMAP soil moisture serves as a general indication of relative soil wetness, as we note at line 396. We will include a discussion of the lack of soil moisture calibration as a study limitation. We did note a seasonal pattern in SMAP soil moisture data, with higher values in the winter. This is physically realistic, as winters in the region are typically rainy with low evapotranspiration but could also include a seasonal bias component. We will also acknowledge this potential limitation in our discussion section, but do not have in-situ data available to verify if it is occurring or not at our sites.*

3. **The current work adopts a previous threshold of 5 mm to distinguish landslide from non-landslide events. While a reference is provided, it would be beneficial to include rationale for this threshold in the current manuscript. From the reviewer's perspective, whether internal movement of 5 mm should be classified as "landslide" is worth discussion. Such movement could represent only localized slope displacement rather than strain bifurcation or connection into a plastic zone. Please justify why the 5 mm threshold is effective for classifying sites as landslide locations.**

*Response to Comment 3: Thank you for this important comment. In the revised manuscript, we will provide a clearer rationale for the use of the 5 mm displacement threshold. As described in our response to Comment 3 from Reviewer #1, the thresholds were selected based on the empirical distribution of displacement changes measured at the top of the slide plane across all inclinometer readings. The cumulative distribution of these displacement changes (Rahimikhameneh et al., 2024) shows that approximately 50% of readings involve less than 1 mm of movement (likely within the instrument's measurement uncertainty) whereas about 13% show displacement changes greater than 5 mm. These values form natural separation points within the dataset. For this reason, we classify ≥ 5 mm of internal*

movement between consecutive readings as a landslide event, and < 1 mm as negligible movement.

It is important to emphasize that the ≥ 5 mm movements used in our classification represent reactivation along pre-existing shear zones, rather than the initiation of new landslides. The inclinometer casings are installed at sites already identified as unstable, and the detected displacements reflect movement along established failure surfaces. Thus, it is a threshold for identifying periods of renewed movement at known unstable highway slopes and not at sites without a history of movement. This clarification will be added to the revised manuscript.

4. **Regarding performance metrics, is it possible within the current research framework to plot a Receiver Operating Characteristic (ROC) curve and compute the Area Under the Curve (AUC)?**

*Response to Comment 4: Thank you for this suggestion. As part of the revision process, we generated ROC curves and computed AUC values for all five thresholds. The ROC curves are shown in Figure R1, which will be included in the revised manuscript. Both NSM-dependent thresholds (A and B) yield higher AUC values than the Marino et al. (2020) and Guzzetti et al. (2008) thresholds. Threshold B achieves the highest AUC among all evaluated methods and shows a slight improvement relative to the Godt et al. (2006) model, while Threshold A exhibits comparable performance.*

5. **The current work uses normalized soil moisture-dependent thresholds. How does this approach affect the uncertainty or sensitivity of predictions across space? Future work could include variogram or Bayesian analysis to investigate spatial uncertainty**.

*Response to Comment 5: Thank you for this comment. In the current study, NSM is used as a regional hydrologic indicator, with the normalization intended to provide a consistent index. We did not assess spatial uncertainty or sensitivity in our approach. We agree that incorporating variograms or Bayesian analysis would be a valuable addition to future studies, and we will explicitly mention these techniques as promising directions for future work.*

6. **Please add references for Pandas, NumPy, OS, and Matplotlib as a way to support the open-source community.**

*Response to Comment 6: Thank you for catching this. In the revised manuscript, we will add formal references for NumPy, Pandas, Matplotlib, and the Python OS module.*

7. **For Figures 2 and 3, consider plotting moving averages to better illustrate seasonal or annual changes in soil moisture.**

*Response to Comment 7:* *Thank you for the helpful suggestion. We have revised Figure 3 to include monthly moving averages of soil moisture rather than raw daily values (Figure R2). Applying a moving average smooths high-frequency noise and filters out short-term fluctuations that are not relevant to capturing antecedent or seasonal moisture conditions.*

[Figure]

**Figure R2. Time series of inclinometer displacement (red stars indicate landslide events and open symbols indicate non-landslide events), daily rainfall (purple bars), and monthly moving average soil moisture (SMAP L4, blue line) for (a) AL-69 Inclinometer 13002 and (b) AL-5 Inclinometer 13002A.**

8. **In Figures 6 and 7, the legend shows the thresholds as shaded blocks, while the plot presents them as lines. Please make these representations consistent.**

*Response to Comment 8:* *Thank you for pointing this out. We will update Figures 6 and 7 to ensure consistency between the legend representation and the plotted threshold lines.*

9. **In Figure 9, the legend notation "1.25 to<2.15" reads awkwardly. Consider using a simpler format such as "1.25–2.15."**

*Response to Comment* 9*: Thank you for this suggestion. We agree that the notation in Figure 9 can be clearer. In the revised version, we will update the label from "1.25 to < 2.15" to "1.25–2.15."*